# When, where, and how to add new neurons to ANNs

**Kaitlin Maile**[1]  **Emmanuel Rachelson**[2]  **Hervé Luga**[1]  **Dennis G. Wilson**[2]

[1]IRIT, University of Toulouse, Toulouse, France
[2]ISAE-SUPAERO, University of Toulouse, Toulouse, France

**Abstract**  Neurogenesis in ANNs is an understudied and difficult problem, even compared to other forms of structural learning like pruning. By decomposing it into triggers and initializations, we introduce a framework for studying the various facets of neurogenesis: when, where, and how to add neurons during the learning process. We present the Neural Orthogonality (NORTH*) suite of neurogenesis strategies, combining layer-wise triggers and initializations based on the orthogonality of activations or weights to dynamically grow performant networks that converge to an efficient size. We evaluate our contributions against other recent neurogenesis works across a variety of supervised learning tasks. [1]

## 1 Introduction

Training deep artificial neural networks (ANNs) usually involves pre-selecting a static architecture and then optimizing the parameters of that architecture for a given dataset and task (Goodfellow et al., 2016). However, the field of *structural learning* aims to optimize both the architecture and its parameters during the learning process through techniques such as neural architecture search (NAS) (Elsken et al., 2019) or pruning (Deng et al., 2020). By dynamically building ANNs throughout learning, we can aim towards networks which learn not only parameters but also their architectures for specific tasks, potentially improving learning or performance of the final architecture and avoiding expensive hand-tuning of architectures. Furthermore, dynamic ANN architectures can allow continual growth to include more information and tasks as necessary. Dynamic architecture changes may add or remove structural units or connections, operating on the basis of neurons, channels, layers, or other structures. We consider neurogenesis to be the form of structural learning that adds new neurons or channels within existing layers, complementary to structural pruning often used by channel-searching NAS techniques such as Li et al. (2017), Wan et al. (2020) and to network deepening such as Wen et al. (2020). Our focus in this paper is neurogenesis, which is a naturally difficult and less well-studied form of structural learning that invokes multiple questions, specifically when to add neurons, where to add them in the network, and how to initialize the parameters of new neurons; questions which have an undefined search space that must be artificially constrained for optimization.

While considering the trade-off of performance versus efficiency of the network, including training time, inference time, and memory size, we aim to grow ANNs without a preset final size. We propose that an effective neurogenesis algorithm should converge to an efficient number of neurons in each layer during the course of training.

We present the following contributions towards understanding and utilizing neurogenesis.

- We introduce a framework for neurogenesis, decomposing it into triggers and initializations.

- We present novel triggers and initializations based on orthogonality of either post-activations or weights that together form the NORTH* suite of strategies, as well as a gradient-based trigger to complement existing gradient-based initializations. These algorithmically answer when, where, and how to add new neurons to ANNs.

---

[1]Our code is available here: `https://github.com/neurogenesisauthors/Neurogenesis`

- We compare these contributions with existing initialization methods from the literature and baselines in a variety of tasks and architecture search spaces, showing that NORTH* strategies achieve compact yet performant architectures.

## 2 Background

### 2.1 Problem Statement and Notations

An artificial neural network (ANN) may be optimized through empirical risk minimization:

$$\arg\min_{f} \mathbb{E}_{\boldsymbol{x},\boldsymbol{y}\sim D}\, L(f(\boldsymbol{x}), \boldsymbol{y}) \tag{1}$$

for a neural network $f$, a loss function $L$, and a dataset $D$ consisting of inputs $x$ and outputs $y$. In the simple case of a dense multi-layer perceptron (MLP) with $d$ hidden layers, $f$ may be expressed as $f(\boldsymbol{x}) = \sigma_{d+1}(\boldsymbol{W}_{d+1}\sigma_d(...\boldsymbol{W}_2\sigma_1(\boldsymbol{W}_1\boldsymbol{x})...))$, where $\sigma_l$ is a nonlinear activation function, adding a row for the bias, and $\boldsymbol{W}_l \in \mathbb{R}^{M_l \times (M_{l-1}+1)}$ is the weight matrix for the $l^{\text{th}}$ layer, including the bias parameters. We distinguish between input and output weights of neurons in $\boldsymbol{W}_l$ with $M_l$ rows that each represent the fan-in weights of a neuron in layer $l$ receiving input from each of the $M_{l-1}$ preceding neurons and equivalently $M_{l-1} + 1$ columns for which each of $M_{l-1}$ are the fan-out weights of a previous-layer neuron and the last column is the biases.

We define $\boldsymbol{z}_l = \boldsymbol{W}_l\sigma_{l-1}(\boldsymbol{W}_{l-1}\sigma_{l-2}(...\boldsymbol{W}_2\sigma_1(\boldsymbol{W}_1\boldsymbol{x})...))$ as the pre-activations and $\boldsymbol{h}_l = \sigma_l(\boldsymbol{z}_l)$ as the post-activations of layer $l$. For $n$ samples, i.e. a mini-batch, $\boldsymbol{Z}_l \in \mathbb{R}^{M_l \times n}$ is the pre-activation matrix and $\boldsymbol{H}_l = \sigma(\boldsymbol{Z}_l)$ is the post-activation matrix.

To perform neurogenesis, the addition of $k$ neurons to the $l^{\text{th}}$ layer is accomplished by appending $k$ rows of fan-in weights to $\boldsymbol{W}_l$ and $k$ columns of fan-out weights to $\boldsymbol{W}_{l+1}$. Utilizing this operation in the empirical risk minimization of Equation (1) defines the neurogenesis optimization. We restrict the search space of where to add new neurons to within existing layers.

### 2.2 Neurogenesis

We consider neurogenesis through the following dimensions.
- When: Standard learning algorithms naturally discretize the learning process into steps, so neurogenesis may occur at any step of training.
- How many: When neurogenesis is triggered at a step, multiple neurons can be added at once.
- Where: In standard ANN architectures, this amounts to which layer neurogenesis occurs in.
- How: The fan-in and fan-out weights and bias of each new neuron must be initialized.

Most existing neurogenesis works focus on "How", or the initialization of neurons, using a fixed schedule of additions. Some works have approached global scheduling, such as Ash (1989); Bengio et al. (2005); Wu et al. (2019) using convergence as a global trigger for neurogenesis. However, this leads to neurogenesis happening later in training and does not explore the questions of "Where" or "How many". We posit that, in order to be competitive with static networks in terms of training cost, neurogenesis should happen throughout training, including before convergence.

To answer "How", most existing non-random neurogenesis initializations in recent literature are gradient-based. Firefly from Wu et al. (2020) generates candidate neurons that either split existing neurons with noise or are completely new and selects those with the highest gradient norm. Du et al. (2019) also create noisy copies, selecting from high-saliency neurons. Both GradMax from Evci et al. (2022) and NeST from Dai et al. (2019) compute an auxiliary gradient of zero-weighted bridging connections across layers over a mini-batch. GradMax uses left singular vectors of this auxiliary gradient as fan-out weights to maximise the gradient norm, while NeST initializes a sparse neuron connecting neuron pairs with a high gradient value in their auxiliary connection.

---

**Algorithm 1** Neurogenesis framework.

---

**procedure** NEUROGENESIS(TRIGGER, INITIALIZATION, initial ANN $f$)
    **while** $f$ not converged **do**
        Gradient descent step on current existing weights
        **for** each hidden layer $l$ **do**
            **if** TRIGGER$(f, l) > 0$ **then**
                Add TRIGGER$(f, l)$ neurons using INITIALIZATION$(f, l)$
    **return** trained $f$

---

No known works add new neurons that are explicitly more different to existing neurons than randomly selected weights. In the context of non-growing and linear networks, Saxe et al. (2014) proposes orthogonal weight initialization. In the case of MLPs with nonlinear activation functions, a layer with orthogonal weights only provides orthogonal pre-activations, and only if the output of the previous layer is orthogonal. Daneshmand et al. (2021) finds pre-activation orthogonality may actually be detrimental in non-linear networks and presents a weight initialization scheme for layers with equal static widths that maximises post-activation orthogonality, but does not have a clear extension to neurogenesis. Mellor et al. (2021) proposes a NAS heuristic based on post-activation distances between samples, computationally similar to the orthogonality gap metric by Daneshmand et al. (2021). Maintaining orthogonality of neuron activations in static networks has especially been studied in reinforcement learning (Lyle et al., 2022) and self-supervised learning (Jing et al., 2022), but becomes relevant in all tasks when considering dynamically growing networks.

Neurogenesis during the learning process necessarily expands the parameter search space and thus the loss landscape's dimensionality. Neurogenesis may either immediately preserve the ANN's function, such as by using zero fan-in or fan-out weights as in Evci et al. (2022); Islam et al. (2009), or may change the function and current performance. Function-preserving neurogenesis is a form of network morphism (Wei et al., 2016; Chen et al., 2016), maintaining the location in the newly expanded loss landscape but changing the direction of the descent path into the new dimensions. Evci et al. (2022) selects the steepest direction by maximising the gradient norm. Function-altering neurons jump from the descent path: Kilcher et al. (2018); Wu et al. (2019, 2020) escape convergence at saddle points, while Dai et al. (2019) performs a backpropagation-like jump.

Previous neurogenesis works take many approaches across a range of motivations and applications to tackle this difficult problem. In order to unify neurogenesis strategies and study their components, we propose a framework within which we formulate our contributions.

## 3 A Framework for Studying Neurogenesis

Our neurogenesis framework decomposes neurogenesis strategies into *triggers*, which are heuristics that determine when, where, and how many neurons to add, and *initializations*, which determine how to set their weights before training them. We present the basic framework in Algorithm 1. After every gradient step, for each layer, the trigger is evaluated to compute if and how many neurons to add, then the initialization is used to add these new neurons. As the trigger is evaluated after each gradient step, the decision of when is based on non-zero outputs from the trigger. Similarly, the question of where to add neurons is treated by evaluating the trigger on each layer independently.

We introduce our contributions for the case of MLPs, although they generalize to more complex cases like convolutions, detailed in Appendix A.4. Our strategies with their triggers and initializations are summarized in Table 1. We define triggers in Section 3.1 and initializations in Section 3.2.

### 3.1 Triggers

We explore three sources of information for triggers for neurogenesis: neural activations, weights, and gradients. These triggers determine if and how many neurons to add for each layer after each

Table 1: Specification of initialization and trigger pairs used for each strategy.

| Strategy | Trigger | | Initialization | |
|---|---|---|---|---|
| NORTH-Select | $T_{act}$ | eq. (3) | Select | sec. 3.2 |
| NORTH-Pre | $T_{act}$ | eq. (3) | Pre-activation | eq. (7) |
| NORTH-Random | $T_{act}$ | eq. (3) | RandomInit | sec. 3.2 |
| NORTH-Weight | $T_{weight}$ | eq. (5) | Weight | eq. (8) |
| GradMax | $T_{grad}$ | eq. (6) | GradMax | sec. A.3, Evci et al. (2022) |
| Firefly | $T_{grad}$ | eq. (6) | Firefly | sec. A.3, Wu et al. (2020) |
| NeST | $T_{grad}$ | eq. (6) | NeST | sec. A.3, Dai et al. (2019) |

gradient step. We propose that a useful neurogenesis trigger should add neurons when doing so will efficiently improve the capacity of the network or learning.

**Activation-based.** When building an efficient network through neurogenesis, we intuitively want to add neurons yielding novel features that may improve the direction of descent, lower asymptotical empirical risk, or avoid unnecessary redundancy in the network. Thus, we need to measure how different or orthogonal the post-activations are from each other. To measure orthogonality of a layer, we use the $\epsilon$-numerical rank of the post-activation matrix (Kumar et al., 2021; Lyle et al., 2022). For layer $l$ and $n$ samples generating the post-activation $H_l$, the effective dimension metric may be estimated by

$$\phi_a^{ED}(f, l) = \frac{1}{M_l} \left| \left\{ \sigma \in \text{SVD}\left(\frac{1}{\sqrt{n}} H_l\right) \middle| \sigma > \epsilon \right\} \right|, \tag{2}$$

where $\text{SVD}\left(\frac{1}{\sqrt{n}} H_l\right)$ is the set of singular values of $\frac{1}{\sqrt{n}} H_l$ and $\epsilon > 0$ is a small threshold. Due to the SVD decomposition, evaluating this metric has a constraint on the dimensions of $H_l$ of $n > M_l$.

When the orthogonality metric of a layer's current post-activations is high, there may be additional useful features beyond the currently saturated directions so we add neurons. Conversely, a low metric value indicates redundancy in the layer, which may benefit from differentiation between neurons via gradient steps before reconsidering neurogenesis. Adding a new neuron will increase the metric when its activation is more orthogonal to that of other neurons and decrease it if it is redundant. We use each layer's metric value at network initialization as the baseline value, which accounts for orthogonality loss as the input is propagated through layers the network. We aim to at least maintain this initial metric value over training, even as the layer grows: if new neurons do not increase the rank and thus lower the metric value, then neurogenesis pauses until the rank increases via gradient descent. We multiply the baseline value by a threshold hyperparameter $\gamma_a$ close to 1. The number of neurons triggered is the difference of the current metric value and the threshold, scaled by the current number of neurons:

$$T_{act}(f, \phi_a, l) = \min\left(0, \lfloor M_l \left(\phi_a(f, l) - \gamma_a \phi_a(f_0, l)\right) \rfloor\right), \tag{3}$$

where $f_0$ is the ANN at initialization and $\phi_a$ is an orthogonality metric.

**Weight-based.** We include weight matrix orthogonality as a comparison to activation-based methods. The trigger, $T_{weight}$, is computed as in equations (2)-(3) but with $W_l$ instead of $H_l$:

$$\phi_w^{ED}(f, l) = \frac{1}{M_l} \left| \left\{ \sigma \in \text{SVD}\left(\frac{1}{\sqrt{n}} W_l\right) \middle| \sigma > \epsilon \right\} \right|, \tag{4}$$

$$T_{weight}(f, \phi_w, l) = \min\left(0, \lfloor M_l \left(\phi_w(f, l) - \gamma_w \phi_w(f_0, l)\right) \rfloor\right). \tag{5}$$

**Gradient-based.** In order to study gradient-based neurogenesis initializations such as Dai et al. (2019), Wu et al. (2020), and Evci et al. (2022) in the context of dynamic neurogenesis, we propose a trigger based on the auxiliary gradient. As shown by Evci et al. (2022), the maximum increase

in gradient norm by adding $k$ neurons to the $l^{\text{th}}$ layer is the sum of the largest $k$ singular values of the auxiliary gradient matrix $\frac{\partial L}{\partial z_{l+1}} h_{l-1}^\intercal$. We use these singular values and compare them to the gradient norms of all existing neurons in that layer over the same $n$ samples. The number of neurons triggered is the number of singular values larger than the sum of gradient norms:

$$T_{grad}\left(f, L, l\right) = \left|\left\{ \sigma \in \text{SVD}\left(\frac{\partial L}{\partial Z_{l+1}} H_{l-1}^\intercal\right) \middle| \sigma > \sum_{m=1}^{M_l} \left\|\frac{\partial L}{\partial w_m^{\text{in}}}\right\|_F + \left\|\frac{\partial L}{\partial w_m^{\text{out}}}\right\|_F \right\}\right|,$$ (6)

where $w_m^{\text{in}}$ is the $m^{\text{th}}$ row of $W_l$ and $w_m^{\text{out}}$ is the $m^{\text{th}}$ column of $W_{l+1}$, respectively representing the fan-in and fan-out weights of the $m^{\text{th}}$ neuron in layer $l$. We intuit this threshold creates neurons that could have initial gradients significantly stronger than existing neurons and will be harder to surpass as the layer grows, thus leading to convergence in layer width.

## 3.2 Initializations

For the initialization of new neurons, we aim to reuse information computed for the corresponding trigger in each method to reduce computational overhead. For all NORTH* initializations, we add function-preserving neurons which do not immediately change the output of the network, or more specifically the activation of any downstream layers. We propose that the role of initialization for neurogenesis during training is rather to add a new neuron that is useful locally to the triggered layer which will then be integrated into the network's functionality through gradient descent; we measure this utility based on activation, weights, and gradients. We rescale all new neurons to match the weight norm of existing neurons, so scaling is ignored in the following calculations for simplicity.

**Activation-based.** We present multiple approaches for initializing neurons towards orthogonal post-activations. Due to the nonlinearity of a neuron, we only have a closed-form solution for a set of fan-in weights yielding not the desired orthogonal post-activation but an orthogonal pre-activation, similar to related works on orthogonality in static networks Saxe et al. (2014). The following approaches generate candidate neurons and select those that independently maximise the orthogonality metric of the post-activations with the existing neurons in that layer. We initialize fan-out weights to 0. See Appendix A.2 for more details.

- Select: generate random candidate neurons.
- Pre-activation: generate candidate neurons with pre-activations maximally orthogonal to existing pre-activations. For $n$ samples and $M_l$ neurons in layer $l$, a candidate's fan-in weights are

$$w = (H_{l-1}^\intercal)^{-1} V_{Z_l}'^\intercal a,$$ (7)

where $(H_{l-1}^\intercal)^{-1}$ is the left inverse of the transpose of post-activations $H_{l-1}$ for layer $l-1$, $V_{Z_l}'$ is comprised of orthogonal vectors of the kernel of pre-activations $Z_l$, and $a \in \mathbb{R}^{n-M_l}$ is a random vector resampled for each candidate.

As a baseline to these selection-based techniques, NORTH-Random uses RandomInit as the initialization strategy with random fan-in weights and zeroed fan-out weights.

**Weight-based.** For the weight-based strategy NORTH-Weight, maximally orthogonal weights for new neurons can be solved for directly. We initialize $k$ neurons each with fan-out weights of 0 and fan-in weights projected onto the kernel of $W_l$:

$$w = \text{proj}_{\text{ker}W_l}(w_i),$$ (8)

where $w_i \in \mathbb{R}^{M_{l-1}+1}$ are the random initial fan-in weights from the base weight initialization and the projection is computed using $V_{W_l}'$, comprised of orthogonal vectors of the kernel.

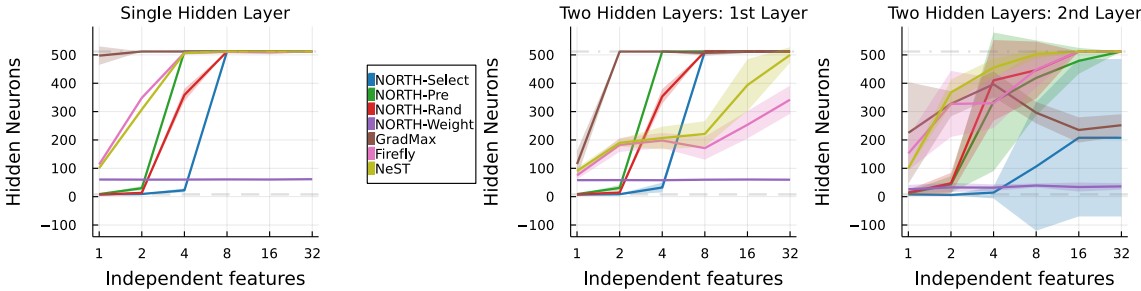

Figure 1: Hidden neurons upon convergence on for MLPs on generated toy data versus the number of independent features in a 1 (left) and 2 (middle, right) hidden layer MLPs.

**Gradient-based.** We reimplement the neurogenesis initialization portions of NeST from Dai et al. (2019), Firefly from Wu et al. (2020), and GradMax from Evci et al. (2022) within our framework to investigate them in the dynamic context and be able to compare them with our dynamic methods. We summarize the strategies and note any differences from the original works in Appendix A.3.

## 4 Experiments

We study the impact of the different trigger and initialization methods detailed in Table 1 over a variety of tasks, with dynamic schedules to study triggers and independently studying the importance of initialization by also using fixed schedules. We also compare with static networks of various sizes to understand the relative performance of networks grown with neurogenesis. Specifically, we focus on three settings: MLPs (Rumelhart et al., 1986) on generated toy datasets in Section 4.1, MLPs on MNIST (Deng, 2012) in Section 4.2, and convolutional neural networks (CNNs) on image classification: VGG-11 (Simonyan and Zisserman, 2015) and WideResNet-28 (Zagoruyko and Komodakis, 2016) on CIFAR10/CIFAR100 in Section 4.3. For all experiments, plots show mean and standard deviation in the form of error bars, clouds, or ellipses (which may result in artifacts outside of the layer size bounds) or quartiles in the form of boxplots across 5 seeded trials. We include further experiment details in Appendix B.1 and a study on hyperparameters in B.2. We only compare our results within our framework rather than take reported results in order to avoid confounding differences in implementation and focus on understanding neurogenesis. By studying neurogenesis across different tasks, architectures, and layer types, we aim to draw general conclusions about when, where, and how to add new neurons to ANNs.

### 4.1 Generated Data

We first analyze all strategies on supervised learning for toy generated binary classification datasets, described in Appendix B.1.1. All have 64 input features but differ in how many of these inputs are linearly independent, demonstrating how neurogenesis responds to the effective dimensionality of the input. We train MLPs with 1 or 2 hidden layers using the different neurogenesis strategies in all layers, shown in Figure 1. In this simple task, all strategies achieve similarly high test accuracy across datasets: above 99% for all feature sizes except 97% for 32 independent features.

Comparing final layer widths shows the dependency of each strategy on the effective input dimensionality and base architecture. The first layers across the two architectures exhibit the unsupervised nature of NORTH* strategies but also the strength of gradient-based strategies to adapt width to depth. Width of the second hidden layer has higher variance across all strategies, indicative of the layer-wise input distribution: the input to the second layer is changing in dimension and distribution, whereas the first hidden layer has a static input from the dataset in this task.

NORTH-Weight is restricted by the input dimensionality due to the upper bound on number of singular values by the minimum of the matrices' dimensions. Thus, it cannot explore networks wider than the input dimensionality, which may be problematic for tasks with low-dimensional

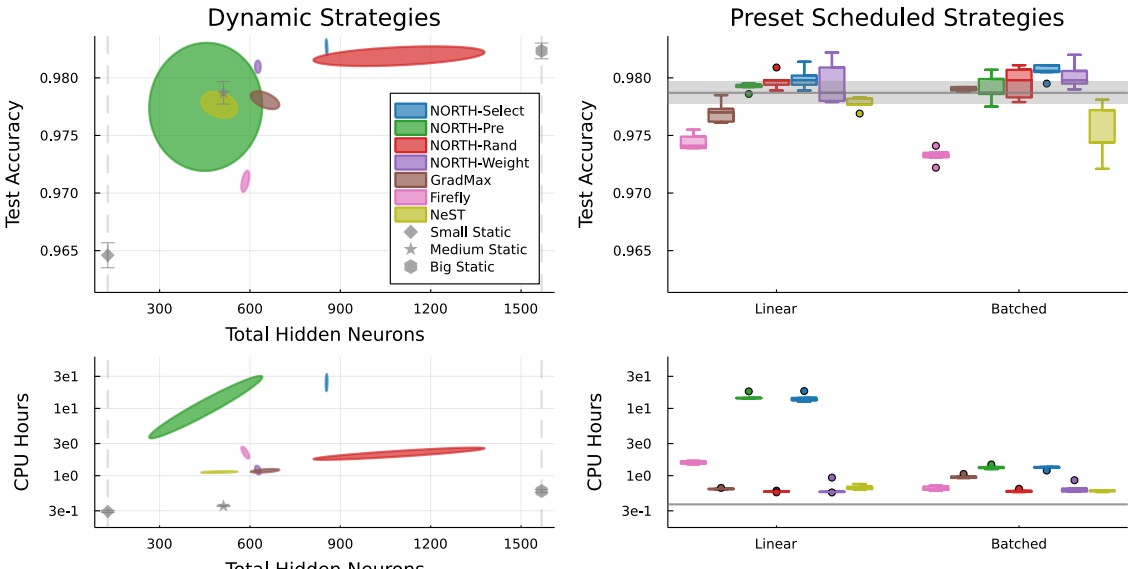

Figure 2: Dynamic (left) and preset scheduled (right) comparisons of network performance across neurogenesis strategies for MLPs on MNIST, evaluated by test accuracy (top) and training time (bottom). Preset scheduled are compared against the Medium Static network, which is the same architecture as the final size of the grown networks.

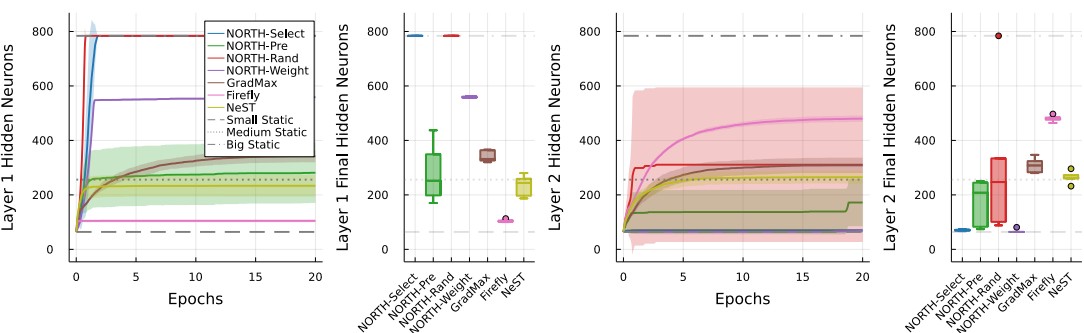

Figure 3: Layer width for MLPs on MNIST over training for dynamic strategies, across training (far-left for Layer 1, center-right for Layer 2) and at the end of training (center-left for Layer 1, far-right for Layer 2).

inputs. All other NORTH* methods grow to the maximum size allowed in the first layer and to larger sizes than NORTH-Weight in the second layer.

## 4.2 MLP MNIST

We continue our study on neurogenesis in dense MLP layers on the task of MNIST classification using networks with vectorized image input, 2 hidden layers which grow using neurogenesis, and 10 output neurons. We test dynamic strategies as well as isolate the initializations in preset growth schedules, which are detailed in Appendix B.1.2. We compare test accuracy and training time against final network size or schedule in Figure 2, detailing the layer widths over training for dynamic schedules in Figure 3.

For dynamic neurogenesis, the Pareto front of accuracy versus network size is dominated by the NORTH* orthogonality-based methods, demonstrating the advantage of using activation and weight orthogonality to trigger neurogenesis.

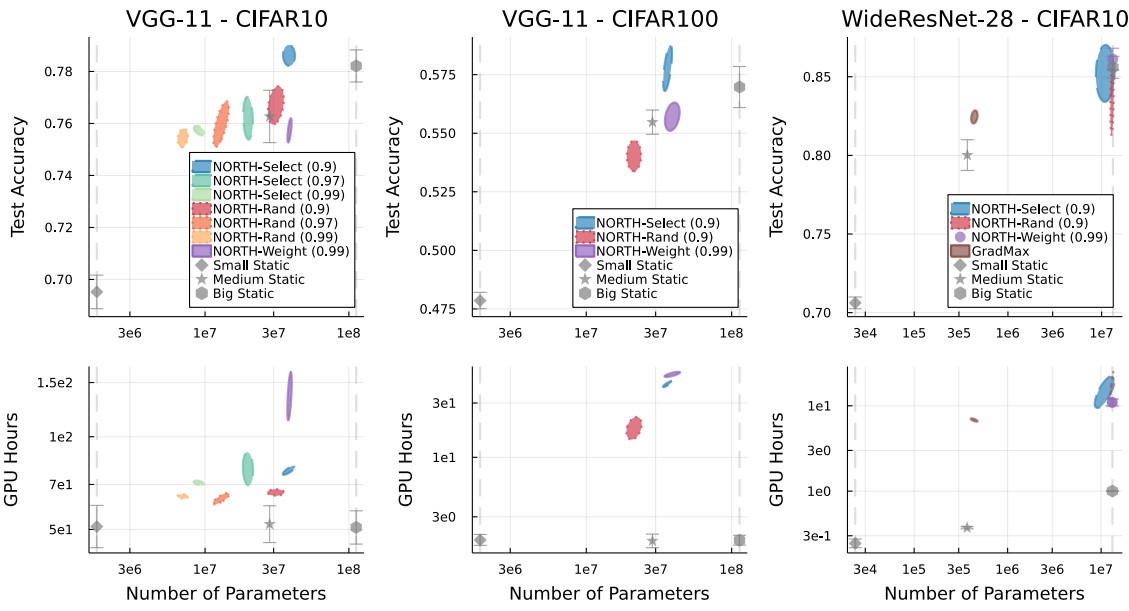

Figure 4: Comparisons of test accuracy (top) and training time (bottom) vs. network size across neurogenesis strategies and values of $\gamma$ for VGG-11 on CIFAR10 (left) and CIFAR100 (center) and WideResNet-28 on CIFAR10 (right).

The gradient-based methods, especially GradMax and NeST, yield comparable but not quite competitive results, possibly confounded by the adaptations used to implement them within our framework. However, they are faster in this CPU implementation than some activation-based NORTH* strategies, benefiting from thorough code optimization of deep learning libraries for gradient descent, while the larger SVD calculations used by NORTH* strategies hinder efficiency.

From Figure 3, no dynamic strategy reached the maximum possible network size. Thus, our triggers do respond to the "When" and "How many" questions, converging towards smaller networks. Furthermore, NORTH-Select and NORTH-Random reach networks which are competitive with the largest non-growing network with fewer neurons. NORTH* methods tend to converge towards networks with large first layers and smaller second layers, a common pattern in hand-designed networks. Thus, the triggers can also respond to the "Where" question by dynamically adding neurons to each layer independently. However, using a maximum layer size is still needed for these methods to avoid exploding layer widths, particularly without extensive hyperparameter tuning.

As for preset schedules, multiple NORTH* strategies exceed the performance of the static network of the same final architecture for this task and hyperparameter setting, demonstrating that dynamically growing a network over learning can result in better performance than starting with a network of the maximum final size. NORTH-Pre slightly under-performed the less-informed approaches of NORTH-Select and NORTH-Random, as well as NORTH-Weight which has a similar premise towards orthogonal pre-activations.

### 4.3 Deep CNNs on CIFAR10/CIFAR100

We extend neurogenesis to deep convolutional networks and test them on CIFAR10 and CIFAR100 classification with a VGG-11 or WideResNet-28 backbone. Due to restricted time and resources, we only test the dynamic NORTH* methods applicable to convolutions against the non-growing baselines; further details are in Appendix B.1.3.

Test accuracy and training cost of the different methods is compared against network size for each base architecture and dataset combination in Figure 4. On VGG-11, our NORTH* methods find networks in the same order of size of the standard network, here represented by Medium Static,

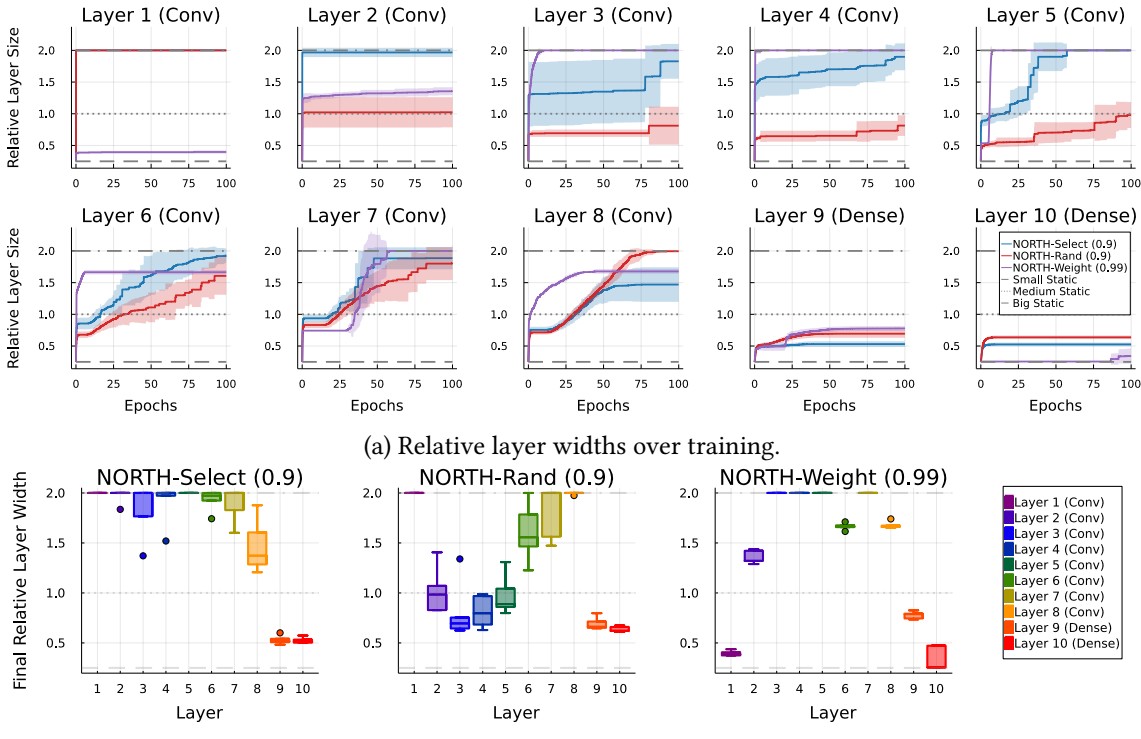

(a) Relative layer widths over training.

(b) Relative layer widths at the end of training.

Figure 5: Layer widths for VGG-11 on CIFAR10, scaled relative to standard VGG-11 widths, for NORTH-Select, NORTH-Rand, and NORTH-Weight, over the course of training in (a) and at the end of training in (b).

while being competitive or improved in accuracy for both datasets. On CIFAR10, we search across $\gamma_a$, showing the tunability of NORTH-Select and NORTH-Rand across the trade-off for size and performance. Notably, NORTH-Select with $\gamma_a = 0.9$ outperforms even Big Static with less than half as many parameters on both CIFAR10 and CIFAR100. As in the dense case, NORTH-Select has a slightly higher cost for the benefit of smarter candidate generation compared to NORTH-Rand. For WideResNet-28, NORTH* methods tended to grow to the at or near the maximum size, likely due to the complicated dynamics of residual connections on the trigger metrics. NORTH-Weight slightly beats Big Static at this maximum size while NORTH-Select matches performance, and GradMax beats Medium Static at a similar size.

Layer width during and at the end of training for VGG-11 on CIFAR10 is presented in Figure 5, showing how different NORTH* methods respond to the "Where" question. All three methods keep the dense layers relatively small, which has a large effect on the overall parameter count. NORTH-Weight grows some intermediate convolutional layers to maximum size, almost opposite to the width patterns of NORTH-Rand. Deeper convolutional layers tend to have later innovations, likely in response to innovations of the earlier layers.

## 5 Discussion

NORTH* strategies grow efficient networks which achieve comparable performance to static architectures, even surpassing the performance of static architectures of the same final size. We posit that the use of growth with informed triggers leads to efficient networks as redundant neurons have a low chance of being created in the small network initialization and are not added by NORTH* neurogenesis. This highlights a benefit of neurogenesis: dynamically creating an architecture adapted to the target task.

We find that activation and weight orthogonality can be useful triggers for dynamic neurogenesis and weight initializations of new neurons. Previous methods use gradient information for these decisions, explicitly considering training dynamics and particularly the outputs via the loss. Our NORTH* methods only implicitly use this information via network updates changing orthogonality metrics and explicitly use the input distribution. This could allow extension to unsupervised and semi-supervised contexts, where gradient information may be unavailable or unreliable.

As neurogenesis triggers are evaluated after every gradient step, they can have significant computational cost. NORTH* strategies were more efficient than gradient-based strategies in our GPU experiments, but less for CPU. The number of candidates, frequency of trigger evaluations, and the size of the activation buffer are hyperparameters of NORTH* which can be reduced for efficiency or increased for performance. Comparing the cost of growing networks to that of static networks is difficult: although dynamically grown networks incur higher training costs than predetermined schedules or non-growing networks, they have the benefit of algorithmically determining the architecture widths instead of hand-engineering them through expensive trial-and-error.

We focus on function-preserving neuron addition. Such neurons do not immediately impact network output while still receiving gradient information to become functional through training. This can be easily achieved by zero-ing either the fan-in weights (as in NORTH*) or the fan-out weights (as in GradMax). Firefly and NeST, however, initialize functional neurons that change the position in the loss landscape; this could hinder performance by moving the network in an undesired direction, but could also complement gradient descent through neuron initialization. Optimizing the fan-out weights of NORTH* strategies to make helpful movements in the loss landscape is a direction for future work.

This work serves as a study to isolate neurogenesis from other structural learning techniques in order to understand it more and open the door to further neurogenesis research. We pair complimentary triggers and activations; other combinations are possible albeit with careful consideration. We show that neurogenesis alone can already find compact yet performant networks. We leave dynamic structural learning, incorporating neurogenesis along other structural operators like layer addition and pruning, as future work. For layer addition, we hypothesize that similar activation and weight-based triggering applies, then layer addition candidates could be compared to neuron addition candidates; for gradient-based methods, evaluating the auxiliary gradient on directly neighboring layers rather than alternating could signal layer addition. As for pruning, we hypothesize that unstructured pruning may not be necessary with our strategies that avoid redundancy at initialization, but structured pruning could remove neurons that become redundant over training. This could bypass the need for hyperparameters such as maximum layer width and be even more effective towards searching for networks that optimize both performance and cost.

**Limitations and Broader Impact Statement**: In addition to the limitations and broader impacts already discussed, the efficient and adaptive architectures grown by neurogenesis have reduced environmental footprint, avoiding cycles of hand-tuning static architectures or training expensive super-networks to prune or apply NAS to. We do not identify any potential negative societal impacts beyond those associated with general-purpose machine learning methodologies.

## 6 Conclusion

We present the NORTH* suite of neurogenesis strategies, comprised of triggers that use orthogonality metrics of either activations or weights to determine when, where, and how many neurons to add and informed initializations that optimize the metrics. NORTH* strategies achieve dynamic neurogenesis, growing effective networks that converge in size and can outperform baseline static networks in compactness or performance. The orthogonality-based neurogenesis methods in NORTH* could be further used to grow networks in a variety of settings, such as continual learning, shifting distributions, or combined synergistically with other structural learning methods. Neurogenesis can grow network architectures that dynamically respond to learning.

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
