## A  Supplementary - Methods

### A.1  Orthogonality Measure

We additionally study a metric based on the orthogonality gap from Daneshmand et al. (2021) that measures the gap of the covariance matrix of post-activations to the identity matrix. We use an estimate of the orthogonality gap

$$\phi_a^{OG}(f,l) = 1 - \left\| \left( \frac{1}{\|H_l\|_F^2} \right) H_l^\top H_l - \left( \frac{1}{\|I_n\|_F^2} \right) \right\|_F, \tag{9}$$

where $I_n \in \mathbb{R}^{n \times n}$ is the identity matrix. $H_l^\top H_l \in \mathbb{R}^{n \times n}$ represents the covariance of $H_l$ across samples and thus approaches the identity matrix if the post-activation vector of each neuron is orthogonal to all others. However, this metric has a dependency on layer width that may confound neurogenesis strategies. Also, this metric does not extend to the weight-based strategy.

### A.2  Activation-based Initializations

For pre-activation based initialization, we compute the Moore-Penrose pseudoinverse $(H_{l-1}^\top)^+$, which approximates the left inverse when $n \gg M_{l-i}$ and the rows of $H_{l-1}$ are approximately orthogonal. The orthogonal vectors of the kernel of pre-activations $Z_l$ are computed as the right $n - M_l$ columns of $V_{Z_l}$ from the full singular value decomposition of $Z_l$.

In addition to the two methods presented for creating candidates for activation-based selection at neuron initialization, we also present an optimization-based approach. We perform projected gradient descent (Bertsekas, 1997) on randomly generated candidate neurons, optimizing the orthogonality metric of each set with the existing neurons and constrained by weight norm.

When the effective dimension metric is used, we need a continuous proxy for selection across all three methods as well as a differentiable proxy for the projected optimization. We use the sum of singular values as the differentiable proxy as well as the tie-breaker for neurons that yield equivalent effective dimensionalities for the layer.

However, the optimization method was found to be prohibitively expensive, performing an inner optimization for every growth event with an additional projection step for every gradient step. Further study into post-activation orthogonality metrics and proxies is warranted, particularly balancing performance with efficiency.

### A.3  Gradient-based Initializations

- NeST from Dai et al. (2019): initialize $k$ neurons, using sums of squares of the largest auxiliary gradient matrix values with a random sign to the respective fan-in and fan-out weights as connections. As NeST is only specified for a single neuron, we adapt to $k$ neurons by using different random signs for each neuron. We use the same recommended top quantile of 0.4 for filtering the largest auxiliary gradient matrix values. In the convolutional case, generate random candidate neurons and add those that independently lower the loss the most.

- Firefly from Wu et al. (2020): generate candidate neurons from two strategies: split existing neurons by halving the fan-out weights and adding and subtracting random uniform noise from the two copies, and add new candidates with random fan-in weights and small fan-out weights. We then keep the $k$ neurons with the largest gradient norm. We do not perform gradient steps on the candidates, as done in Firefly, in order to focus on neurogenesis rather than pruning after training; Firefly does both. The magnitude of the noise for split candidates and the fan-out weights for new candidates is controlled by hyperparameter $\epsilon_{\text{Firefly}}$, for which we use a value of 1e-4.

- GradMax from Evci et al. (2022): initialize $k$ neurons with zero fan-in weights and the top $k$ left singular vectors of the auxiliary gradient matrix as the fan-out weights. We note that this method is limited to $M_{l+1}$ candidate neurons, capping neurogenesis at the size of the next layer.

Table 2: Default hyperparameters.

| | MLP - Generated | MLP - MNIST | VGG-11 - CIFAR10 | VGG-11 - CIFAR100 | WRN-28 - CIFAR10 |
|---|---|---|---|---|---|
| Optimizer | ADAM (Kingma and Ba, 2015) | | | ADAM, CosineAnnealing | |
| Learning Rate | 3e-4 | | | | 3e-3 |
| Batchsize | 128 | 512 | 128 | | |
| Epochs | Convergence | 20 | 100 | | 50 |
| Input dimensions | 64 | 784 | $32 \times 32 \times 3$ | | |
| Output dimension | 2 | 10 | 10 | 100 | 10 |
| Hidden layers/groups | 1 or 2 | 2 | 10 | | 4 |
| Initial layer width | 4 | 64 | $0.25\times$ | | |
| Medium Static width | N/A | 256 | $1\times$ | | |
| Maximum layer width | 512 | 784 | $2\times$ | | $6\times$ |
| Buffer size | 1024 | 1568 | 128 | | |
| Base initialization | Xavier uniform (Glorot and Bengio, 2010) | | | | |
| Orthogonality metric | $\phi_{ED}$ | | | | |
| $\epsilon$, eq. (2),(4) | 0.01 | | | | |
| $\gamma_a$, eq. (3) | 0.97 | | 0.9, 0.97, 0.99 | 0.9 | |
| $\gamma_w$, eq. (5) | 0.99 | | | | |
| Device | CPU | | GPU | | |

## A.4 Neurogenesis for Convolutions

For convolutional layers, we consider that a channel is analogous to a neuron in a dense layer. However, the activation for a channel and a single sample as well as the parameterization of the connection between two channels are both matrices instead of single values. Thus, $H_l \in \mathbb{R}^{H_l \times W_l \times M_l \times n}$ and $W_l \in \mathbb{R}^{k_l \times k_l \times M_l \times (M_{l-1}+1)}$. We detail how our methods are adapted to convolutions as follows.

- Effective Dimension: $H_l$ is flattened to $\mathbb{R}^{M_l \times (H_l W_l n)}$, so that the metric is relative to the number of neurons in layer $l$. This permits the use of a smaller buffer size.

- Orthogonality Gap: $H_l$ is flattened to $\mathbb{R}^{(H_l W_l M_l) \times n}$, so that $H_l^\top H_l \in \mathbb{R}^{n \times n}$.

- Activation-based Trigger: Because the orthogonality metric at initialization is very close to 0 for the last layers of a deep convolutional network and increases as the network learns, we instead use the running maximum of orthogonality metric values for the threshold. Thus,

$$T_{act}^{conv}\left(f, \phi_a, l\right) = \min\left(0, \left\lfloor M_l \left(\phi_a\left(f, l\right) - \gamma_a \max_t \phi_a\left(f_t, l\right)\right)\right\rfloor\right). \tag{10}$$

In the case of residual networks, we turn off residual connections when evaluating activation-based metrics to isolate the contribution by each non-identity layer. We also grow groups of layers with interdependent channel constraints due to these residual connects based on metrics of the first layer of the group.

- Weight-based Trigger and Initialization: $W_l$ is flattened to $\mathbb{R}^{M_l \times ((M_{l-1}+1)k_l k_l)}$.

- Gradient-based Trigger: While we did not have enough resources to include results for all dynamic gradient-based neurogenesis strategies in convolutional neural networks, the implementation for these are included in our code. Computing the auxiliary gradient is detailed in Evci et al. (2022).

- Activation-based Initialization: We do not have a closed-form solution for orthogonal pre-activations of convolutions, so we do not implement NORTH-Pre for architectures with convolutions.

## B Supplementary - Experiments

### B.1 Experiment Details

We run 5 trials with random seeds 1-5 in each study for each configuration and present aggregated results. We list hyperparameters in Table 2. For initializations that generate candidates, the number of candidates is set to either 1000 for dense layers or 100 for convolutional layers, plus the number of neurons to add. For all initializations, the bias is initially set to 0 and non-zero fan-in weights and fan-out weights of new neurons are scaled to match the current weight norm of existing neurons in the same layer for all initializations, except for Firefly for which we scale noise vectors to $\epsilon_{\text{Firefly}}$ = 1e-4 times the current weight norm of existing neurons. We use a modified ReLU activation function as GradMax requires that the gradient at 0 is 1; in other words, $\sigma(x) = 0$ if $x < 0$ and $\sigma(x) = x$ if $x \geq 0$, resulting in the same activation but a different gradient at 0 compared with standard ReLU $\sigma(x) = \max(0, x)$. We noted no empirical difference in results between the standard and modified ReLU and thus use the modified ReLU across all configurations (Bertoin et al., 2021).

We compare growing networks against static networks: Small Static is the same size as the initial size of growing networks, Medium Static is the same size as the final size for preset schedules and is also the baseline used for preset schedules, and Big Static is the same size as the maximum size of growing networks.

All plots except 5 average across the 5 random seeds and use standard deviation for error bars, clouds, or ellipses. The ellipses represent the contour line of a Gaussian density function matching the mean and covariance at 1 standard deviation.

We ran all CPU experiments on Intel Xeon Gold 6136 computing nodes and the GPU experiments on GTX 1080TIs, RTX8000s, and Tesla V100s, all provided by local clusters. The experiments to generate all plots and tables in this work consumed 299.53 CPU days and 85.17 GPU days in total, with an estimated carbon footprint of 236.3 kgCO$_2$e (Lacoste et al., 2019; Lannelongue et al., 2021).

### B.1.1 Simulated Data.
The datasets are created for a given number of independent features $n_f$ by generating 5000 samples of $n_f$ normally distributed features and filling the remaining $64 - n_f$ features with random linear combinations. The output binary class for each sample is generated by summing all features and adding 10% random noise, then thresholding by the mean value. 10% of samples are reserved as the test split for final accuracy.

### B.1.2 MLPs on MNIST.
The $28 \times 28$ images of the MNIST dataset are flattened to 784 features. This is fed into an MLP with 2 hidden layers, then classified into 10 classes representing the digit in the image. We use the standard training and test split.

To better understand the contribution of the trigger versus initialization, we compare initialization methods using fixed neurogenesis schedules. We use two predetermined schedules: Linear, which adds one neuron per gradient step, and Batched, which adds batches of neurons after each gradient step. Both schedules are defined by initial and final layer widths. We grow neurons for the first 75% of epochs. The linear schedule adds neurons one at a time so it has many small growth events, while the batched schedule adds neurons in 8 regular intervals so it has only a few large growth events. We compare with a Medium Static network which starts training with the final size of the growing networks, 256 neurons per hidden layer.

### B.1.3 VGG-11 and WideResNet-28 on CIFAR10 and CIFAR100.
We use the standard training and test splits of CIFAR10 and CIFAR100, which have input images of size $32 \times 32 \times 3$ and 10 or 100 classes, respectively. The standard VGG-11 hidden layer widths are 64, 128, 256, 256, 512, 512, 512, and 512 channels in the convolutional layers, then 4096 and 4096 neurons in the dense layers. The standard WideResNet-28-1 has an initial convolution layer with 16 channels, 3 groups with 16, 32, and 64 channels each and 4 residual blocks per group, and a final dense layer connecting global mean pooling to the output. The backbone used as the starting point for neurogenesis has $\frac{1}{4}\times$

Table 3: Average GPU seconds per mini-batch for trigger evaluation for all layers, including GPU memory garbage collection time, during the 1st epoch of VGG-11 training on CIFAR10 on a Tesla V100.

| Strategy | Time |
|----------|------|
| NORTH-Select | $5.97 \pm 0.18$ |
| NORTH-Rand | $5.77 \pm 0.09$ |
| NORTH-Weight | $10.05 \pm 1.01$ |
| GradMax | $14.00 \pm 7.03$ |
| Firefly | $18.52 \pm 11.59$ |
| NeST | $23.79 \pm 8.78$ |
| Small Static | $5.49 \pm 0.05$ |
| Medium Static | $5.49 \pm 0.05$ |
| Big Static | $5.14 \pm 0.05$ |

the standard layer widths, also used for the Small Static architecture. The layer sizes for growing networks were limited to 2× the standard layer widths for VGG-11 and 6× the standard layer widths for WideResNet-28-1 (which is WideResNet-28-6 in their notation), also used for the Big Static architecture. hlDue to the interdependence of layer widths in WideResNet-28, we measure the first layer of each group to determine when, where, and how to add neurons to all layers of the same group. We also turn off the residual connections when evaluating activation and weight-based metrics. We use batch-normalization in WideResNet-28, but otherwise do not perform any other "tricks" such as data augmentation or dropout. Due to restricted computational time, we could not perform a formal sweep of hyperparameters. We selected the largest learning rate that we tested that allowed all networks up to Big Static to increase accuracy during the first 5 epochs. Similarly, we selected the largest batchsize that we tested that could fit any gradient computation in memory on all GPU devices. The cluster GPUs available to us were not identical, so we ran each base architecture and dataset combination on the same GPU in order to compare within experiments, but not across.

The gradient-based strategies require computing auxiliary gradients in addition to the gradients of existing parameters and any gradients further required in gradient-based initialization, shown in Table 3. Because batchsizes are generally selected to maximise GPU memory of each gradient evaluation and the gradient-based neurogenesis methods use multiple passes per batch, they are less efficient than NORTH* in the case of VGG-11.

## B.2  Hyperparameter studies on MLPs for MNIST

In the following experiments, we study the effect of NORTH* hyperparameters independently in the MNIST task. These experiments were used to inform our default settings for the experiments in 4.2. We discuss our findings in the context of this specific task and setup. We only use the standard training split of MNIST in these experiments, holding out 10% of this split as the validation set to measure final validation accuracy.

We study the effects of activation orthogonality measures Effective Dimension $\phi_a^{ED}$ from Equation 2 versus Orthogonality Gap $\phi_a^{OG}$ from Equation 9 on performance and training time and their sensitivities to $\gamma_a$ in Figure 6. $\phi_a^{ED}$ generally yields higher accuracies given the network size, although $\phi_a^{OG}$ is generally more efficient. $\phi_a^{ED}$ is more robust to varying $\gamma_a$ and yields less extreme network sizes, although both metrics are rather sensitive to its value. Thus, applying activation-based neurogenesis to new tasks may require tuning of $\gamma_a$ to the specific task configuration. $\phi_a^{ED}$ has the additional hard constraint on sample size being greater than the current layer width, which may make scaling to very wide networks difficult.

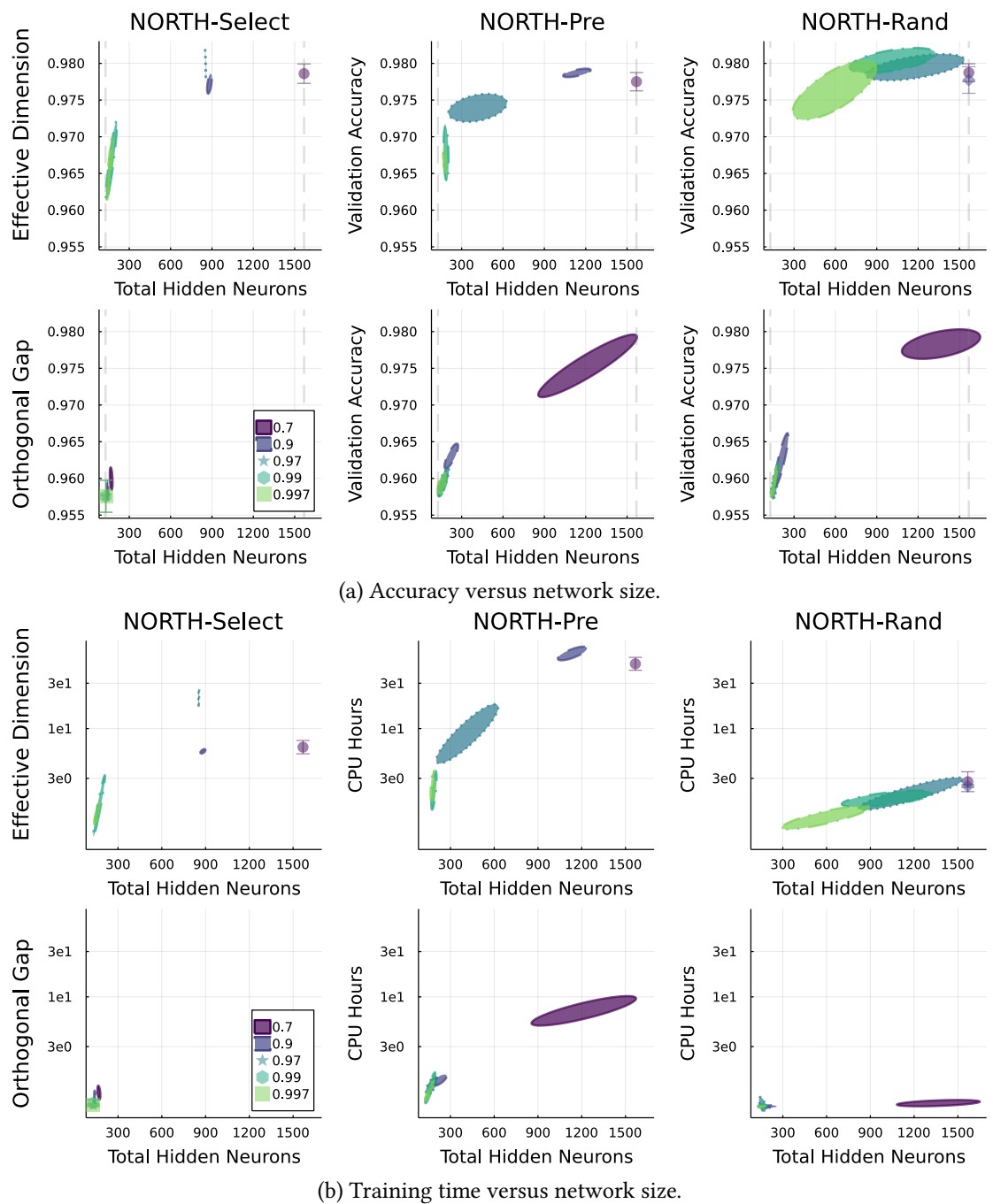

(a) Accuracy versus network size.

(b) Training time versus network size.

Figure 6: Comparison of validation performance and sensitivity to $\gamma_a$ of $\phi_a^{ED}$ and $\phi_a^{OG}$ in activation-based strategies.

We study the effect of batchsize in Figure 7a. We note that the size of the buffer used for assessing the orthogonality metric is independent of batch size. The dynamic networks generally benefit more from larger batchsizes than static networks. NORTH-Select particularly matches or exceeds the validation performance of larger static networks at larger batchsizes. We hypothesize that the relative frequency of neurogenesis trigger and initialization steps for larger batchsizes is beneficial.

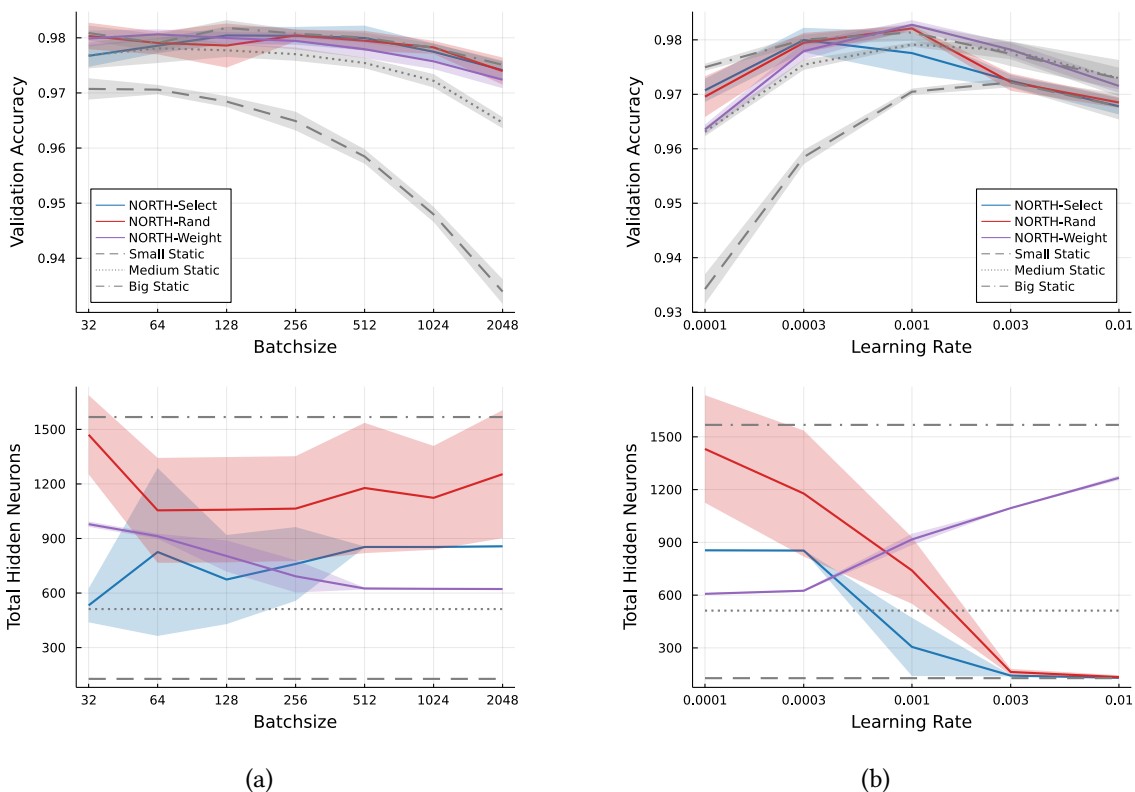

Figure 7: Effects of (a) batchsize and (b) learning rate on NORTH-Select and NORTH-Random, compared to dynamic and static baselines.

The results of various learning rates are shown in Figure 7b. The general trend is that larger networks, both static and growing, benefit from smaller learning rates. NORTH-Select matches NORTH-Random in validation performance at smaller learning rates, even with a smaller network size. We conjecture that larger learning rates cause large, noisy gradient steps that hinder the activation-based triggering of neurogenesis while increasing randomness and thus orthogonality for weight-based triggering.