# OpenReview forum: "When, where, and how to add new neurons to ANNs"
_automl.cc/AutoML/2022/Track/Main — AutoML-Conf 2022 (Main Track)_

### Official Review · Reviewer_4ixA · 2022-03-23

**Potential Impact On The Field Of Automl Rating:** 2
**Technical Quality And Correctness Rating:** 2
**Clarity Rating:** 3

**Summary Of Contributions:**

The presented paper is situated in the NAS literature, particularly in dynamically learned architectures; i.e. the network is grown during the training phase alongside the optimization of weights. The authors try to answer the question of where, when and how to add neurons to layers to increase the capacity of the network midst training. They do this by introducing layer-wise triggers and initialization strategies. Crucially, they use three sources of information for their triggers; neural activations, weights and gradients, which help identify where and how many neurons should be added. Essentially, their heuristics all build on the orthogonality measure of Lyle et al. 22's layerwise heuristic, i.e. counting the nonsingular values of some weight-related matrix, where some threshold decides whether the singular value is counted or not. The first two approaches, namely activation-based and weight-based, both define the number of added neurons in reference to the ANN at its initialization (which, from my understanding, supposedly is a running baseline over the training process). The gradient-based flavor does not require this heuristic baseline but instead uses the in- & output weights respective loss gradient norms.

Regarding how to add the new neurons, the authors decided to add them in a function-preserving manner. The subsequent gradient steps inform the new neurons' contribution. Again, the authors present three initialization techniques: activation-, weight- and gradient-based. The activation-based entails two flavors, both of which try to find initialization for the neurons towards the post-activations, where no analytical solution exists. The weight-based version initializes via the orthogonal kernel of the current layers' weights. Lastly, the gradient-based approach simply re-implements existing work.


**Clarity:**

f_0 should be detailed more explicitly. It is difficult to understand whether or not this is a running baseline or a fixed one.

The subsubchapters are sometimes hardly longer than three lines. A separate subchapter should convey a standalone or subsequent argument of considerable size. The present distinction presents the flavors as completely distinguishable entities. But considering that they are mostly variations of the same metric, they rather should be presented in the context of their unity and consequently in paragraphs of the same chapter.

Eq. (7)'s random distribution should be clarified. Further, please motivate why the left inverse is used.

Figure 2: Top right; the gray bar across all seems not appropriately motivated in the text.

Figure 3: Their methods hit the maximally available neurons (big static) early on, such that the sentence in line 227 is incorrect (at least for the first layer).

Figure 5: This is the least informative; plotting the seeds separately hardly indicates anything interpretable.

Table 2: It is poorly formatted. It is unclear, to which column some of the entries refer.

Eqs. (2) & (4) and (3) & (5) hardly deserve to stand on their own. Instead, I recommend to write one abstract equation each, where the differences are based on function inputs. This will improve readability and highlight the differences. Further, this will give space to highlight the motivation for those equations, which the authors should consider.

Please elaborate on the motivation regarding eq. 6.


**Overall Review:**

General topic:
The overall idea of neurogenesis is timely and a highly relevant area of research to arrive at smaller and or sparser models that require less time to train and are sufficiently capable. From a theoretical point of view, growing architectures on their own without the ability of a contraction is troubling by the authors' own argument - since the addition of neurons might be dependent on the location in the loss surface. In particular, this could imply that later evolutions might find vast amounts of redundancy, which could suggest that smaller models are favorable again. The nonlinear dependencies between layers may aggravate this fact.

The theory's equations can easily be compressed into 4 "major" equations (2, 3, 6, 7), where eq. 2 & 6 are insufficiently motivated adaptations of Lyle et al.'s orthogonality metric. Regarding novelty, apart from the subsequent hyperparameter analysis, these are their core theoretical contributions, which they should emphasize and explain more intricately.

Regarding the hyperparameters:
The sensitivity to the selection thresholds & hyperparameters epsilon and gamma are apparent based on the equations already - and not surprisingly, the authors suggest optimizing them (explicitly gamma) for each new task. Depending on the invested computation, this might partially defeat the purpose of a more lightweight training procedure. On their own, the hyperparameter's sensitivity suggests that this method is primarily a heuristic - just as the models they compare against. However, the dynamic perspective inherent in their evidence-driven heuristic significantly improves over static methods conceptually.

Initialization:
Finding initializations that are not function-altering in the context of nonlinearities is a tedious task except for zeroing their outputs. The pre-activation procedure is interesting but should be motivated better.

The figures will definitely require some clarification.


**Potential Impact On The Field Of Automl:**

NAS allows finding suitably capable and promising architectures for a given task. Neurogenesis is a particular flavor that grows architectures bit by bit and during training, implying it could find smaller networks without the necessity of training a plethora of architectures until convergence. The promise neurogenesis holds is that it seeks to find small architectures efficiently during the training process of the ANN. Their approach could reduce the computational burden significantly.

The paper's impact is more subtle. It experiments with a slight deviation of the orthogonality metric proposed by Lyle et al. 22’ in the construction of a layer's features' orthogonality. The rationale is that low orthogonality in a layer indicates redundancy and thus should stunt growth. In contrast, high orthogonality should foster growth since there might be more to learn if only more dimensions were available. Based on this assumption alone, the growth is heuristically determined and has to be parameterized with the hyperparameters epsilon and gamma. The baseline network f_0 and the gradient-based flavor are built on intuition only and lacks a proper motivation beyond trying out several ideas. A success of this paper would primarily necessitate research in the reasoning of these hyperparameters and heuristics before resulting in definite rationals and recommendations.

An important unaddressed issue is that only width is considered, i.e. the number of layers (depth) is held constant. This fact alone restricts the applicability, as it ignores the possibility of an increasingly non-linear latent feature space. Nevertheless, given a fixed number of layers, it rightly answers the allocation problem of neurons. Another issue is that the continuous learning problem is allowed only to dynamically grow, but not to shrink - which, depending on the placement on the loss surface could be a reasonable thing to do by their own metrics (since a lack of orthogonality would imply redundancy). This issue is also empirically shown by their experiments. An expand-shrink scheme would yield an unconstrained and truly dynamic growth procedure.



**Reproducibility:**

The source code has no documentation or comments regarding any functionality, which the authors affirmed they did. This would result in a significant struggle when interpreting or applying the codebase. Since understanding the implementations - especially of third party's models- is crucial to evaluating the relative performance, reproducing the code would be a fairly involved problem.

Attributed to the aforementioned fact, it is hard to make out where the hyperparameter optimization procedure occurs. A pointer would be very much appreciated.

Regardless, I would expect that the provided scripts will reproduce the results and plots deterministically. However, it is difficult to assess how those results are aligned with their claims due to the lack of documentation.


**Review Confidence:**

4: You are confident in your assessment, but not absolutely certain. It is unlikely, but not impossible, that you did not understand some parts of the submission or that you are unfamiliar with some pieces of related work.

**Review Rating:**

2: Reject, not good enough

**Review Summary:**

Overall, the research area in which this paper is situated is an interesting and promising one.
The paper's novelty is relatively limited and requires more elaborate motivation or even experiments supporting their intuition. Summary statistics on the layer's orthogonality metric over the course of optimization would be very much appreciated to support their argument.
The experimental results exhibit apparent issues and the proposed heuristics seem to foster an almost immediate increase in the number of neurons which is constant thereafter. This is only weakly aligned with the method's claim of dynamicity.


**Technical Quality And Correctness:**

The drawn conclusions seem to be correct given their experiments. The visualizations supporting their argument are troubling though;

Figure 1:
1. Why is a negative number of neurons possible?
2. How can the maximum number of neurons (500) be exceeded as indicated by the bounds?
It would help to offset the two experiments here: single layer & two hidden layers. They should either be in different figures or visually offset to clarify that they are not immediately comparable.

Figure 2:
1: The claim of Pareto domination is not necessarily supported by a plotted Pareto front.

Figure 3:
1. How and why could an approach be allowed to surpass the maximum number of neurons?
2. Why does "select" and "rand" almost instantly hit the maximum allowed number of neurons? If this maximum is hand-crafted, it most certainly would be interesting to see what the actual converged number of neurons looks like. Otherwise, these models only effectively differ in the subsequent layer from the big-static model.
3.  In the right plot: how is it possible to have a negative number of neurons in the red bounds?
4. Why would the models Select, Rand, Weight and firefly exhibit almost no variability?

Figure 4:
The plot reads as follows: The NORTH approaches outperform small static and are comparable with medium static. However, they are outperformed by the big static - significantly one might add - based on the repetitions. So it appears there still is some trade-off between growing a network and training the full-sized flavor. Despite lacking in performance, the models are indeed significantly smaller.

Figure 5:
The cumulative rainbow-colored plot is not particularly useful. Layer-wise boxplots across the seeds are much more sensible to overview the algorithms' preferences. Further, y’s metric suggests at first glance that the found models are roughly 8 to 13 times wider than the VGG-11 - albeit a detailed investigation reveals that this is not the case. In a layer-wise relative-layer width boxplot across seeds, the original layer width should be marked as 1. Whether or not the boxplot is below or above this threshold would be visually clearer.

Most troubling is Figure 3, which seems to suggest that the proposed models
a) tend to reach for the maximum capacity (and were manually bounded (?))
b) only grow initially and maintain the size thereafter, defeating the continuous learning premise. This, in my humble opinion, should be the crucial goal that neurogenesis should strive to achieve.

---

### Official Review · Reviewer_2Kre · 2022-04-04

**Potential Impact On The Field Of Automl Rating:** 3
**Technical Quality And Correctness Rating:** 3
**Clarity Rating:** 4

**Summary Of Contributions:**

>>>
Motivation:

The focus of this paper is on neuro-genesis -- the study of automatically expanding neural networks for hosting more knowledge during training.

>>>
Background:

This paper can be seen as an extension of  Lyle et al.'s 2021 work on understanding network capacity. In their paper, Lyle et al. proposed a metric based on singular value decomposition (SVD) to quantify the efficient usage of network capacity during reinforcement learning (RL). This helped Lyle et al. to open the black box and understand how the RL agent was interacting with the environment.

In this neuro-genesis paper, the authors showed that the SVD could be used in a different way. They showed that the SVD metric could serve as an indicator of the capacity of learning; and when all eigenvalues are large, this is a signal that more neurons should be added into the network in order to host more knowledge.

**Clarity:**

>>>
Clarity (+):

The clarity of this paper is extremely high.

The abstracts, introduction, and dicussion were strong; the methods section is correct; and the appendix provides a lot of additional information highlighting the details of training.

**Overall Review:**

>>>
Pros (+):

1) Nicely written, well motivated, and good discussion.
2) Adapted an existing technique from RL and gave it a fresh interpretation for neuro-genesis.
3) An lot of experiments covering most of the important aspects.
4) Demonstrated the flexibility of the network.

>>>
Cons (-):

1) A bit lower on novelty (the RL technique was adapted from Lyle et al.)
2) The experiements were slightly simpler and it would be good to see some results of NORTH applied to an architecture including residual connections.

**Potential Impact On The Field Of Automl:**

>>>
Major Contribution (+):

Though the novelty of this paper is slightly low (the method, after all, was adapted from Lyle et al. 2021), the authors found a fresh interpretation for an existing technique in RL and demonstrated its usefulness for meta-learning and automated machine learning, hence bridging the gap between the three fields.

The authors also showed the fexlibility of their neural orthogonality (NORTH) technique. This technique could be applied in a various different levels (on activations, on weights, and on gradients; see pages 3 and 4) of the network to help its users in understanding the behaviour of the backbone network.

In addition, the authors of this paper also conducted a very rigious series of experimental setup and provided the elaborated details of the hyperparameter settings in their appendix. See more in the "Reproducibility" Section.

A thorough analysis was also given to communicate the affects of NORTH on different levels of network processing during training (see pages 8 and 9).

**Reproducibility:**

>>>
Reproducibility (+):

I am very confident that this paper is reproducible.

The code is provided in an anonymised GitHub (though the code is in Julia, a bit of a surprise) and all of the training details can be found in the appendix (for instance, see Table 2 in page 15). The authors also provided the extra information of the carbon footprints (see page 16).

**Review Confidence:**

4: You are confident in your assessment, but not absolutely certain. It is unlikely, but not impossible, that you did not understand some parts of the submission or that you are unfamiliar with some pieces of related work.

**Review Rating:**

5: Accept, good paper

**Review Summary:**

>>>
Final Review:

This paper is nicely written with many nice experimental comparisons and results. The novelty is slightly low because the authors took the existing technique from Lyle et al. However, they gave the existing method a fresh interpretation and demonstrated its usefulness beyond the original purpose.

The experiements could be conducted on more difficult datasets -- for instance on CUB200 with ResNeXt50. However, most of the important aspects are covered.

**Technical Quality And Correctness:**

>>>
Correctness (+):

I am very confidence in the correctness of this study.

>>>
Technical Quality (+):

In addition, the technical quality of this paper was also extremely high. As mentioned in the "Potential Impact On The Field Of AutoML" Section, not only did the authors demonstrated that their method was extremely flexible, they also compared the performances of their own techniques applied at different levels.

Furthermore, the authors also compared their neuro-genesis algorithm to neural networks with fixed architectures of different sizes. They showed that their neuro-genesis algorithm was able to achieve comparable performances while being able to start with an arbitrary size.

More importantly, the authors also compared their own technique with prior work and showed that their technique was able to scale nicely.

>>>
Shortcomings (-):

A slight complaint is on the simplicity of the experiments. The experiments were conducted on the relatively simple MNIST and CIfar10 datasets. The backbone of their MNSIT dataset was a MLP, and it was a VGG-11 for Cifar10.

I checked the GradMax neuro-genesis paper by Evci et al. (2021) (also cited by the authors), and Evci et al's experiments were arguably more extensive.
They conducted 3 datasets over 3 architectures:
1) CIFAR-10 was processed with WRN-28-1 and also with VGG11,
2) CIFAR-100 was processed with WRN-28-1, and
3) ImageNet was processed with Mobilenet-V1.

While I think it is not necessary for the authors to experiment on larger datasets, I think it might be interesting to see NORTH demonstrated on a network with residual connections. This is because residual connections could be found in most if not all of the SoTA architectural designs (e.g., Transformers).

>>>
Additional Comments:

An additional comment that does not affect my score is that, NORTH could potentially be useful for continual learning as well.

---

### Official Review · Reviewer_N7SE · 2022-04-04

**Potential Impact On The Field Of Automl Rating:** 3
**Technical Quality And Correctness Rating:** 3
**Clarity Rating:** 3

**Summary Of Contributions:**

The concentration of the paper is on "neurogenesis" a subfield of structural learning concentrating on improving performance by dynamically growing the neural network via adding neurons to the architecture.
The paper's contributions to the case of MLPs are as follows:
   •	Proposing a framework for decomposing the neurogenesis problem into two steps;
        o	Layer-wised triggers; to identify when, where, and how to add neurons, and
        o	Layer-wised initialization; to initialize the newly added neurons' weight.
   •	Implementing triggers and initializations methods based on orthogonality of post-activation, and weights. Also, a gradient-based trigger is presented for the existing gradient-based initialization methods in the literature.
   •	Then, the orthogonality methods are compared with the existing gradient-based methods.
The paper aims to grow the neural architecture during training without having a predefined final size, and the objective function is to choose the neural network with the minimum loss function value.

**Clarity:**

There can be some clarifications in some parts which are mentioned below:
-	I could not find any information about how much more expensive the method is in comparison to others which is an important concern in practice.
-	In line 127, the mentioned hard constraint can be more discussed since it is not clear why this constraint exists due to the SVD decomposition.
-	In line 129, how do you consider an orthogonality metric is high or low to decide whether the layer needs new neurons or differentiation between neurons should occur? Is there a threshold?
-	In the plots, use legends somewhere that can be related to all the plots, currently, it is located on one of the plots. Also, the colors are hard to differentiate.


**Overall Review:**

The paper certainly covers an interesting and potentially high-impact topic, and it is well written and easy to follow. The results should be reproducible due to the availability of code, data, and documentation of the code. However, it seems like this is a preliminary paper with a new idea but not fully developed yet, and it can be improved and investigated. With reference to the title, it does not seem to me that the authors answered the mentioned question. The evaluation should be more thorough since there is no comparison of the method to the state-of-art methods in the literature that people cite often. So, it is difficult to determine how well it works relative to those methods. Also, it would be more valuable if the authors mentioned how much more expensive their approach is compared to others since this is a major concern in NAS papers. If the comparison between the NORTH* methods and the gradient-based method exist in the CIFAR10 result, we could conclude that this works for the deeper convolutional neural networks as well. Also, the conclusion can be stronger if the mentioned points are addressed since, currently, it is not strong enough compared to the goal set out earlier in the paper.

**Potential Impact On The Field Of Automl:**

The paper is fairly important in the Neural Architecture Search field if it is more developed in the evaluation and conclusion section.

**Reproducibility:**

Regarding the reproducibility of the paper, the main scripts, the generated data, and plot generating scripts are included in the GitHub repository mentioned by the authors. Also, the Github repository contains documentation about how to use the codes, so the paper should be fairly reproducible by any researcher.

**Review Confidence:**

4: You are confident in your assessment, but not absolutely certain. It is unlikely, but not impossible, that you did not understand some parts of the submission or that you are unfamiliar with some pieces of related work.

**Review Rating:**

5: Accept, good paper

**Review Summary:**

Overall, the paper needs some improvements in terms of clarity, evaluation, and conclusion as mentioned in the above sections.

**Technical Quality And Correctness:**

The proposed approach is innovative, and the formulation of the problem is clear; however, more experiments are needed to draw the mentioned conclusion. In the CIFAR10 results, I wish I could see the comparison of the proposed method and the previous gradient-based methods since this will answer the question of whether the proposed approach will work for deeper networks. Also, more NAS approaches can be used as the baseline for comparison, and they are not mentioned/compared in the paper. It is important to mention how much more expensive the technique is compared to mentioned methods since this is a major concern in NAS papers.

---

### Official Review · Reviewer_iNyp · 2022-04-05

**Potential Impact On The Field Of Automl Rating:** 2
**Technical Quality And Correctness Rating:** 3
**Clarity Rating:** 2

**Summary Of Contributions:**

The paper addresses a framework for neural archtecture search (NAS), called neurogenesis: A (small) network can be extended on-the-fly (i.e. while learning) if so-called triggers deliver positive (non-zero) values. The triggers are mainly threshold-driven, measuring the orthogonality of e.g. weights. The ideas (including the initialization of the new weights/neurons) are based on the papers by Lyle et al. (2022) and Evci et al. (2022). It is not fully clear how the present work distinguishes from these papers. Experiments show that almost the same performance (of static large networks) can be achieved with considerably smaller size.

**Clarity:**

It is not clear how the extension and initialization is realized formally. The description is purely verbal and (in my optinion) a little bit nibulous. This should be presented in a formal manner.

**Overall Review:**

PRO:
The framework presented demonstrates on few data sets that this method leads to considerably smaller networks with on a small decay of performance.

CON:
The assessment of the method is based on two data sets only. Therefore I have only little confidence on the claims.
I am missing a comparison with the concept of pruning: Can we get also a similarly small network from a given big one by repeated pruning?


**Potential Impact On The Field Of Automl:**

NAS is certainly an important field of AutoML. The present paper presents a good starting point, but I would not assess it as an elaborated and finished framework. Moreover, since the numerical results are based on two data sets only, my confidence is limited.
Nevertheless, the contributions are some extension of previous work which is worth a citation for work that addresses similar research.

**Reproducibility:**

The results should be reproducible with the software provided on github. I did not test it.

**Review Confidence:**

3: You are fairly confident in your assessment. It is possible that you did not understand some parts of the submission or that you are unfamiliar with some pieces of related work.

**Review Rating:**

4: Marginally above the acceptance threshold (use sparsely)

**Review Summary:**

The contributions of the paper are in a preliminary state. Therefore I have mixed feelings about a positive recommendation.

**Technical Quality And Correctness:**

I do not see major flaws. My concern is the lack of clarity (see below).

Some issues:
- The figures are too small (especially the labeling). The colors in the figures are sometime hardly distinguisable (especially fig. 3).
- The bibliographic data of the references are incomplete. For almost all conference papers the page numbers are missing. There are only authors, year and title for Kilcher et al. (2018). I don't understand why we need an arXiv preprint to cite a work on the Adam optimizer (Kingma & Ba). I guess their ICLR paper from 2015 would be a more solid choice.
See Proceedings of the 3rd International Conference on Learning Representations (ICLR)
- The claims are based on 2 data sets only.

---

### Official Review · Reviewer_kgnK · 2022-04-09

**Potential Impact On The Field Of Automl Rating:** 3
**Technical Quality And Correctness Rating:** 3
**Clarity Rating:** 3

**Summary Of Contributions:**

In this work, the authors present a framework to ensure the systematic use of neurogenesis. Neurogenesis refers to the process of integrating new neurons in the neural network during the learning process. They decomposed the process into two stages – triggers and initializations. Triggers are like heuristics that determine when, where, and how many neurons to add while initializations decide how to set their initial weights. They present a collection of such triggering and initialization strategies based on the orthogonality of activations or weights. They evaluate their approaches against other neurogenesis strategies on three datasets – a generated toy dataset, MNIST dataset, and CIFAR-10 - using two types of networks – MLP and CNN. Their approach generates models that perform competitively.

**Clarity:**

Overall, the paper is well written and organized. It would be great if you can add some details for the general readers about how to interpret the results presented in the form of cloud and ellipse.

Line 229, “Furthermore, NORTH-Select and NORTH-Random reach networks which are competitive with the largest non-growing network with fewer neurons” is not clear to me. Can you please elaborate on that?

**Overall Review:**

Positive:

Generally, neural network models are hand-designed and fixed in size. This work presents a framework that includes a collection of neurogenesis strategies. Therefore, the framework can be used for any task to dynamically grow the network to an efficient size while maintaining “reasonable” performance.

The paper poses the key questions for neurogenesis - when to add neurons, where to add them in the network, and how to initialize the parameters of new neurons. I liked the way they attempted to answer these questions. To me decomposing the process into two distinct stages - triggers and initializations - is a proper direction to achieve modularity in the framework. This will also allow the users to independently choose the best strategies for both stages.


Negative:

In my opinion, the novelty of this work is limited. The presented strategies are mostly based on previous work. They mainly unified the neurogenesis strategies under a framework and analyzed their individual components.

Based on the results presented in Figures 2 and 4, I do not see a clear advantage of the presented strategies compared to the performance of the big static models. The only benefit is the reduced number of parameters. Still, in Figure 2, the CPU Hours vs Total Hidden Neuron plot shows that the CPU Hour for the big model is still lower than the NORTH* strategies. Then what is the benefit of a reduced network size if that takes longer to train?

**Potential Impact On The Field Of Automl:**

This paper presents a suite of orthogonality-based neurogenesis strategies. They have demonstrated the applicability of their neurogenesis strategies to both Multi-Layer Perceptron (MLP) and CNN. This will clearly serve as an important framework for future researchers who want to dynamically grow their network for multi-task learning, meta-learning, or continual learning setting.

**Reproducibility:**

They have filled out the reproducibility list and provided code for reproduction. They also provided some outputs. It would be great to add the instructions for running the experiment on the CIFAR-10 dataset as well.

**Review Confidence:**

3: You are fairly confident in your assessment. It is possible that you did not understand some parts of the submission or that you are unfamiliar with some pieces of related work.

**Review Rating:**

4: Marginally above the acceptance threshold (use sparsely)

**Review Summary:**

The work deals with an interesting issue of automatically adding neurons to a network based on utility. They empirically demonstrate that orthogonality-based neurogenesis strategies can be used to design a competitive architecture. However, in my opinion, the novelty and the performance improvement of the work are not significant, rather it is acceptable.

**Technical Quality And Correctness:**

The experiments are well structured and exhaustive (3 datasets and 2 types of models). The result concludes that their generated models dominate the Pareto front of accuracy versus network size. However, the novelty of the work seems very incremental. They rely mostly on the existing methodology to design the heuristics.

---

### Official Review · Reviewer_fkZz · 2022-04-10

**Potential Impact On The Field Of Automl:** n/a
**Potential Impact On The Field Of Automl Rating:** 1
**Technical Quality And Correctness:** n/a
**Technical Quality And Correctness Rating:** 3
**Clarity:** n/a
**Clarity Rating:** 3

**Summary Of Contributions:**

n/a

**Overall Review:**

n/a

**Reproducibility:**

The authors provide a julia code base which can be very easily installed.
I created a conda reproduction environment with julia>=1.6 and followed the provided instructions to create the package.
A replication of the two provided examples running *runneurogenesis.jl* with names *simtrial* and *mnisttrial* executed without any dependency issues or followup errors in 1h15mins and 17mins, respectively.
The replicatability can be considered as very good.

**Review Confidence:**

4: You are confident in your assessment, but not absolutely certain. It is unlikely, but not impossible, that you did not understand some parts of the submission or that you are unfamiliar with some pieces of related work.

**Review Rating:**

5: Accept, good paper

**Review Summary:**

n/a

---

### Meta-Review · Area_Chair_T2o9 · 2022-04-30

**Recommendation:** Accept
**Confidence:** 4

**Metareview:**

There is a lot of enthusiasm for this paper, but also one dissenting voice. The discussion was productive and the paper improved as a result. Reviewer 4ixA further suggests an experiment that would validate the results, and the authors should consider it. However, even without it, the paper makes a worthwhile contribution to the conference.

---

### Decision · Program_Chairs · 2022-05-13

Accept